# Using Computational Drug-Gene Analysis to Identify Novel Therapeutic Candidates for Retinal Neuroprotection

**DOI:** 10.3390/ijms232012648

**Published:** 2022-10-21

**Authors:** Edward Xie, Urooba Nadeem, Bingqing Xie, Mark D’Souza, Dinanath Sulakhe, Dimitra Skondra

**Affiliations:** 1Chicago Medical School at Rosalind, Franklin University of Medicine and Science, Chicago, IL 60064, USA; 2Department of Pathology, University of Chicago, Chicago, IL 60637, USA; 3Department of Medicine, University of Chicago, Chicago, IL 60637, USA; 4Duchossois Family Institute, University of Chicago, Chicago, IL 60637, USA; 5Department of Ophthalmology and Visual Science, University of Chicago, Chicago, IL 60637, USA

**Keywords:** neuroprotection, retina, drug-repurposing, bioinformatics

## Abstract

Retinal cell death is responsible for irreversible vision loss in many retinal disorders. No commercially approved treatments are currently available to attenuate retinal cell loss and preserve vision. We seek to identify chemicals/drugs with thoroughly-studied biological functions that possess neuroprotective effects in the retina using a computational bioinformatics approach. We queried the National Center for Biotechnology Information (NCBI) to identify genes associated with retinal neuroprotection. Enrichment analysis was performed using ToppGene to identify compounds related to the identified genes. This analysis constructs a Pharmacome from multiple drug-gene interaction databases to predict compounds with statistically significant associations to genes involved in retinal neuroprotection. Compounds with known deleterious effects (e.g., asbestos, ethanol) or with no clinical indications (e.g., paraquat, ozone) were manually filtered. We identified numerous drug/chemical classes associated to multiple genes implicated in retinal neuroprotection using a systematic computational approach. Anti-diabetics, lipid-lowering medicines, and antioxidants are among the treatments anticipated by this analysis, and many of these drugs could be readily repurposed for retinal neuroprotection. Our technique serves as an unbiased tool that can be utilized in the future to lead focused preclinical and clinical investigations for complex processes such as neuroprotection, as well as a wide range of other ocular pathologies.

## 1. Introduction

The irreversible death of photoreceptors and retinal cells such as ganglion cells, RPE in ocular disorders such as diabetic retinopathy (DR), age-related macular degeneration (AMD), inherited retinal dystrophies, and glaucoma causes progressive vision loss and eventual blindness. Knowledge is evolving about how retinal cells die and how cell death is induced and executed, but still no interventions exist to target how retinal cells can be protected in order to save retinal function. The development of successful neuroprotective strategies requires detailed knowledge of the molecular events during retinal degeneration. Neuroprotection preserves functionality and form in neurons damaged by disease, age, oxidative stress, or toxic compounds [1]. Currently, there are no commercially available treatments to prevent retinal cell loss and preserve vision, and there is a critical, unmet need for neuroprotective modalities to improve retinal survival in a multitude of retinal disorders. The conventional neuroprotective strategy relies on pharmacologic agents which target survival pathways or resolve inflammation and oxidative stress [2]. Today, the AREDS drugs from the Age-Related Eye Disease Study (AREDS) studies are the only FDA-approved agents for an AMD, this trial established that antioxidant micronutrients may have neuroprotective effects on photoreceptors and the retinal pigment epithelium (RPE) [3]. Recent literature has reported on many other neuroprotective agents with beneficial effects on the retina that have not yet undergone preclinical or clinical trials [4,5]. Further improving the understanding of the shared pathways and genetics behind retinal diseases may help discover other novel neuroprotective agents that can promote retinal cell survival.

In this study, we use an unbiased, systems biology computational approach to predict new therapeutic targets for retinal neuroprotection using data from the National Center for Biotechnology Information (NCBI) database. For a complex, multifactorial process such as neuroprotection, there is no singular, absolute process or mechanism that can be targeted, and so multiple biochemical pathways must be studied and addressed. A systems medicine approach can use computational and bioinformatics models to integrate information from multiple causal pathways to discern how separate disease processes intersect [6,7]. The models that are built with this approach more accurately depict how gene alterations within a large molecular system can ultimately cause pathology [8]. In the past, this systems medicine method identified potential therapeutic targets for other complex diseases such as refractory epilepsy, various glioma subtypes, asthma, colorectal cancer, AMD, DR, and Alzheimer’s disease [9,10,11,12,13,14,15,16].

Conceiving and developing a single new drug for clinical applications is an expensive and time-consuming process, but network medicine can identify previously approved drugs to repurpose them for novel indications [17]. Drug repurposing is increasingly popularized as a cheaper and more efficient method to identify new drugs for clinical trials [18,19]. While traditional drug discovery on average requires 10–15 years and US$2–3 billion to develop a new drug, it takes 6.5 years and US$300 million on average to take a repurposed drug to market, as the Phase I and II clinical trials have already been previously performed [20]. To our knowledge, no prior efforts have been made to predict potential drugs and chemicals for neuroprotection via a systems medicine computational approach. Using a network-centric method, from all the described genes in retinal neuroprotection to date, we hypothesized that we could identify novel compounds and known drugs for neuroprotective indications.

## 2. Results

Our query of PubMed Gene identified 117 unique genes associated with neuroprotection in the retina. Out of 77,146 candidate drug/compounds in the drug database, our enrichment analysis of the selected retinal neuroprotection genes generated 14,984 chemical compounds against the drug-gene database with an FDR adjusted *p*-value cutoff of 0.05; however, this number includes redundancies suggested by separate databases used by the ToppGene program. We manually selected the top 25 relevant chemical compounds for the final list of neuroprotective genes by filtering out redundancies as well as known deleterious compounds such as particulate matter, H_2_O_2_, ozone, and asbestos (Table 1).

The predicted drug classes with possible benefits based on the relevant genes for retinal neuroprotection include antioxidants (N-Acetylcysteine, ascorbic acid, glutathione, alpha-lipoic acid, melatonin, coenzyme Q10), polyphenol micronutrients (curcumin, apigenin, pterostilbene, naringin), anti-diabetic agent metformin, lipid-lower agents (simvastatin and atorvastatin), and cardiovascular agents (nifedipine, losartan, enalapril). By directly scavenging for reactive oxygen species (ROS), antioxidants thus also show an anti-inflammatory effect by directly removing a significant source of tissue damage. We identified N-Acetylcysteine (*p* = 3.67 × 10^−56^), an antioxidant that directly replenishes glutathione reserves, as the compound with the greatest significance associated with retinal neuroprotection affecting 51 of the 117 neuroprotection genes. Many of the polyphenol micronutrients identified, such as curcumin (*p* = 1.66 × 10^−51^), exhibit an anti-inflammatory effect by attenuating TNF-alpha via the regulation of NF-KB, as well as an antioxidant effect [21]. Curcumin is the most significant polyphenol nutrient identified, affecting 58 of the 117 genes. Other predicted top compounds that are currently used for FDA-approved indications that could be readily repurposed for neuroprotection include metformin (*p* = 1.83 × 10^−29^), simvastatin (*p* = 2.72 × 10^−53^), and nifedipine (8.36 × 10^−33^), which affect 31, 53, and 24 of the 117 genes, respectively. Metformin is the most used drug for type II diabetes that lowers glucose by stimulating insulin sensitivity and inhibiting gluconeogenesis. Meanwhile, simvastatin and atorvastatin are HMG-CoA reductase inhibitors, which is the most used drug class for hypercholesterolemia that directly inhibits cholesterol synthesis. Nifedipine, losartan, and enalapril are drugs used ubiquitously for hypertension and belong to the calcium channel blocker, angiotensin II receptor blocker, and angiotensin-converting enzyme (ACE) inhibitor classes, respectively. Finally, investigational compounds including MEK inhibitors (U0126) and the MAP kinase inhibitor (SB203580) also correlate with neuroprotection-relevant genes. While these experimental compounds demonstrate significant toxicities and are poorly understood, inhibition of MAPK is known to suppress production of inflammatory cytokines and is currently being tested in cancer trials [22].

Several key genes with significance to neuroprotection are commonly affected by the top compounds identified by our analyses. Key genes with high closeness and betweenness centrality in this network include SOD2, TP53, NFE2L2, NOS2, BCL2, BAX, TNF, CASP3, TGFB, AKT1, and IL6, which were all highly associated with hub compounds identified in the gene-drug pathway (Figure 1). Interestingly, all of the aforementioned genes are associated with “regulation of apoptotic processes” per their relevant GO terms for biological processes (Table 2). Meanwhile, genes SOD2, TP53, NFE2L2, BCL2, BAX, TNF, CASP3, TGFB, AKT1, and IL6 are specifically associated with the terms “neuron death”, “response to wounding”, and “aging”, whereas SOD2 and NFE2L2 encode directly for cytoprotective proteins against oxidative stress and inflammation, which suggest a neuroprotective role with modulation of these genes.

Genes identified from the NCBI query were also analyzed with Lynx to identify gene-pathway networks with relations to retinal neuroprotection (Figure 2). The analysis identified key networks including “PI3k-Akt signaling pathway”, “AGE-RAGE signaling pathway in diabetic complications”, ‘EGFR tyrosine kinase inhibitor resistance”, “HIF-1 signaling pathway”, “Brain-derived neurotrophic factor signaling pathway”, “Neuroinflammation and glutamatergic signaling”, and “Spinal cord injury”. Of note, AKT1, which is a key mediator of growth factor-induced neuronal survival, is identified as the gene with the highest closeness centrality in the network [23]. Thus, targeting of these genes and pathways by pharmaceutical agents imply a possible neuroprotective effect in the retina as well.

The comprehensive gene list, drug list, genes correlative for each drug, and number of hits in the genome are available in Appendix A. This data will be publicly available when the manuscript is accepted for publication.

## 3. Discussion

Retinal neurodegenerative diseases cause an irreversible loss of retinal cells, resulting in vision loss as the disease progresses. As biology systems are complex entanglements of closely related elements, any disease process reflects a disruption of multiple elements in a carefully structured system. While an effective neuroprotective strategy to preserve degenerating retinal neurons is urgently needed, there is currently a lack of knowledge or solutions to address this broad, heterogeneous topic. A possible approach is to use network medicine to bridge the gaps in our understanding. Network medicine can reveal the molecular connections across multiple pathways to elucidate biological relationships that cannot be easily understood by conventional reductionist methods of discovery. With the recent advances in multi-omics, it has become possible to understand large-scale biological networks which can be exploited to help us unravel complex processes such as neuroprotection. A computational approach to predict interactions between drugs and the affected genes is much more effortless and economical compared to traditional experimental assays [6]. This approach allows us to bypass the process of studying every unique retinal disease which has multifactorial pathogenesis and to instead observe only the key players and effectors in retinal neuroprotection. In this study, we used a network approach to manifest relationships between the known genes of retinal neuroprotection and existing pharmacologic compounds to predict the best drug candidates. Contrary to a reductionist methodology, our method employs a holistic approach that involves merging numerous pathways of interacting molecular and cellular components. This integrative approach is perhaps a better match for the human disease process.

Our study found many biologically active chemicals that target the genes involved in retinal neuroprotection, including antioxidants, statins, and anti-diabetics such as metformin; however, compounds with antioxidative effects comprised the majority of the top 25 filtered compounds. The retina is particularly susceptible to oxidative stress due to elevated oxygen consumption, a greater proportion of polyunsaturated fatty acids, and persistent exposure to visible light [24]. Oxidative stress driven by the formation of reactive oxygen species [ROS] is strongly associated with not just AMD, but also retinal ganglion cell diseases such as glaucoma [25] and diabetic retinopathy [26]. Photoreceptors cells and the RPE, in particular, serve as the major sites of superoxide generation in the retina to produce the oxidative stress present in DR and AMD [27,28]. The success of the original AREDS study demonstrated the efficacy of antioxidant therapy and nutritional modulation in neuroprotection3. Our findings similarly reflect the importance of anti-oxidative effects in ophthalmic disorders and neuroprotection as antioxidants such as N-Acetylcysteine (NAC), ascorbic acid, glutathione, alpha lipoic acid (ALA), melatonin, Coenzyme Q (CoQ), and various flavonoids all strongly correlate with risk-genes for retinal neuroprotection (Table 1).

The antioxidants in our list are ROS scavengers that can readily cross the blood–brain barrier due to high lipophilicity, and even Vitamin C, a water-soluble compound, can readily cross via the GLUT-1 transporter [29]. They affect several common pathways, including the inhibition of NF-KB activation by TNF, regulation of Nrf-2ARE signaling, antiapoptotic effects, and inhibition of the MAPK. Compounds such as NAC and ALA have demonstrated efficacy in DR and glaucoma animal models and currently also have several ongoing clinical trials for the treatment of neurodegenerative disorders such as Parkinson’s disease [30,31,32,33]. Both have also shown protection against oxidative damage in human RPE cells in vitro [30,34] and inhibition of DR development in animal models [31,35,36]. In a mice model of retinitis pigmentosa (RP), a retinal disease characterized by the degeneration of rod and cone photoreceptor cells, oral and topical applications of NAC improved cone cell survival and function [37]. The treatment of RP with NAC is now currently in a Phase III clinical study after human studies of oral NAC demonstrated improvements in macular cones and tolerability for RP patients [38]. CoQ is another powerful antioxidant that serves as an essential cofactor for the electron transport chain. The supplementation of CoQ is safe and currently recommended by the Canadian Headache Society guidelines for migraine prophylaxis, and it is also widely used for cardiovascular diseases and diabetes, albeit with mixed evidence [39]. Experimental studies in animal models currently indicate that CoQ has protective effects against glaucomatous neurodegeneration and diabetic retinopathy as well [40]. Unfortunately, antioxidants have had limited effectiveness in the past due to challenges with bioavailability for ocular tissue [6]. Since the pathogenesis of degenerative retinal disorders remains tangled and murky, recent clinical studies focus on combined antioxidant therapy (CAT), which combines multiple safe, affordable compounds to address different mechanisms of stress [41]. For example, while ALA supplementation on its own failed to prevent diabetic macular edema or improve visual acuity for type 2 diabetic patients, ALA supplementation combined with genistein, Vitamin C, E, and B did show protective effects for diabetic patients upon electroretinogram analysis [42,43]. While the efficacy of the AREDS formulation is limited to only slowing progression for intermediate and advanced dry AMD patients, they still nonetheless demonstrate the potential value of CAT. Interestingly, the AREDS formulation exerted a similar effect in DR patients as while therapy failed to change visual acuity, it did slow progression over a five-year period [44]. Antioxidants remain a viable option as neuroprotectants, and as new techniques of ocular drug administration such as nanoformulation become available, the use of antioxidants to prevent disease and supplement current pharmacological treatments can be further studied [45]. Ultimately, more research is needed to define the antioxidants and formulations effective against retinal neurodegeneration.

Our findings have also reflected numerous well-known plant-derived polyphenols that relate to genes involved in retinal neuroprotection. These compounds behave as nutritional antioxidants that have demonstrated additional anti-amyloid and anti-inflammatory effects on the brain in early animal studies. In the retina, curcumin, apigenin, pterostilbene, and naringin reduce pro-inflammatory biomarkers in animal models through the inhibition of MAPK and NF-KB pathways and uphold neuroprotective effects [21,45,46,47]. For example, the supplementation of curcumin, the active compound in turmeric, has demonstrated retinal neuroprotection in light-induced retinal degeneration rat models and decreased proinflammatory markers and retinal cell death after oxidative stress [48,49]. Naringin, a major flavonoid found in citrus fruits, also exert antioxidative and retinal neuroprotective effects in animal studies with the suppression of neuroinflammatory markers, as well as improvements in ganglion cell number and cell layer thickness following streptozocin injections [50]. Apigenin, a flavonoid in fruits/vegetables, injected intravitreally, also reduced inflammatory cytokines in the retina and suppressed microglia activation in vivo [51]. Unfortunately, most nutritional drugs end up with weaker results in human trials due to poor bioavailability and ineffective drug delivery. Curcumin has been described for years in the context of retinal pathologies and neuroprotection, but clinical trials have shown that even a high dose (12,000 mg/day) does not escalate serum levels of curcumin, and so recent efforts with curcumin application have been made to combat this issue of bioavailability (Lao et al., 2006; Yang et al., 2021). For example, a prodrug approach has successfully used curcumin diethyl disuccinate [CurDD] to protect RPE cells from oxidative stress-induced death and to decrease H2O2-induced ROS production [52]. Moreover, attempts at creating novel formulations of curcumin such as Norflo (curcumin-phosphatidylcholine) have also demonstrated effectiveness in clinical trials for eye pathologies such as uveitis and central serous chorioretinopathy. Longvida Curcumin, another new formulation, is currently being tested in dry AMD patients for drusen regression [53,54]. To discover the specific composition of antioxidants and nutrients that are useful for retinal neuroprotection, more research is needed on both medication delivery technologies and mathematical correlations for chemical structures and biological activities.

Several FDA-approved drugs that may be readily repurposed, including metformin and various statins, also appeared among the top 25 most significant compounds related to genes associated with neuroprotection. Metformin is the most widely used drug for type II diabetes with known protective properties against many other senescence-related pathologies such as cardiovascular disease, various cancers, and neurodegenerative diseases such as Parkinson’s Disease and dementia [55,56,57]. Previous studies have shown that metformin’s effects on the visual system include inhibition of angiogenesis, decreased vasodilation in retinal blood vessels, and blunting of other inflammatory responses [58,59]. A connection between hyperglycemia and neurodegeneration has been proposed through the action of microglial activity and neuroinflammation, which is why it is hypothesized that anti-diabetic drugs such as metformin may have a neuroprotective effect [60]. The exact connection between hyperglycemia and neurodegeneration is unclear, but diabetic retinopathy as we currently understand it is defined by the interplay of inflammation, neurodegeneration, and vascular alterations in the retina [61]. In animal models of AMD and retinitis pigmentosa, metformin usage helps maintain the integrity of photoreceptors and the RPE via activation of the AMP-activated protein kinase (AMPK) system [62]. Recently, our team and others similarly found evidence that patients taking metformin are associated with a decreased risk of developing retina disorders including AMD and non-proliferative DR when adjusted for age, gender, and other comorbidities [63,64]. It is believed that metformin demonstrates neuroprotection against glutamate-induced excitotoxicity, which is seen in neurodegenerative disorders such as glaucoma and diabetic retinopathy by promoting retinal neuronal cell survival via the MEK/ERK signaling pathway [65,66,67,68]. The exact function of antidiabetics in translational, preclinical, and clinical investigations to explore their possible role in novel therapeutic strategies for neuroprotection needs to be pursued.

Statins are HMG-CoA reductase inhibitors and another class of FDA-approved drugs currently used for lipid-lowering therapy but have also demonstrated neuroprotective effects through various mechanisms including anti-inflammation, apoptosis regulation, and antioxidation [69]. Cholesterol is required for neuronal processes that include neurite formation, myelination, and synapse activity, which means that the brain possesses an incredibly high cholesterol demand as it utilizes around 25% of the total body cholesterol despite only representing 2% of the total body weight [70,71]. In the retina, statins have long been theorized to treat neurodegenerative diseases such as AMD due to the role of cholesterol in drusen formation, a well-known feature of AMD. A recent pharmacokinetic study in humans elucidated how statins can cross the blood–retina barrier especially for the more lipophilic compounds such as simvastatin and atorvastatin which were found in our analysis [72]. One clinical trial reported regression of drusen deposits and vision gain in dry AMD patients receiving high atorvastatin treatment [73]. Outside of AMD, statins have also been shown to exert anti-inflammatory and neuroprotective properties in the eyes of patients with retinal detachment, diabetes, and neurodegenerative posterior segment disorders as well [74,75,76]. In one retrospective cohort study, statin therapy was associated with a lower prevalence of DR and found to decrease the progression of DR in patients with dyslipidemia and T2DM [77]. Our findings thus add to the growing body of evidence that FDA-approved lipid-lowering treatments have the potential to slow or reverse the course of retinal disorders such as AMD and DR. Unfortunately, the relationship between statin therapy and glaucoma remains unclear. While many studies suggest a null relation, a recent 10-year longitudinal study studying long-term statin effects suggests that patients with prolonged statin therapy for >3 years had an increased risk of glaucoma onset; however, this pertained specifically to rosuvastatin, which was not identified in our analysis [78].

Other classes of drugs demonstrated by this analysis include MAPK inhibitors and MEK 1/2 inhibitors. Both the novel MAPK and MEK 1/2 drugs can cause serious ocular toxicities, including retinal vein occlusion, uveitis, and retinal pigment epithelial detachment, which limit their use in the current state [22]. However, these drugs are attractive for AMD as they target the MAPK pathway, which closely intertwines with VEGF and HIF-1 signaling [79]. The work on these drugs is in its infancy and additional experiments to modify the structure and toxicities spectrum of these agents are needed. 

Our present data may also justify new findings of oxidative stress occurring outside the retinal mitochondria. While oxidative stress has long been cited to a play a role in the pathogenesis of degenerative retinal pathologies such as AMD and diabetic retinopathy, traditional knowledge points to retinal mitochondria as the primary target of free radical production [80,81,82]. However, recent studies in vitro have shown an additional element of ectopic oxidative stress outside retinal mitochondria and inside the photoreceptor outer segments (OS), which are modified cilium containing membranous disks that express proteins specialized for oxidative phosphorylation and phototransduction [83]. Furthermore, this machinery of proteins in the OS designed for ectopic oxidative phosphorylation is correlated with increased ATP demand due to photo-transduction. It is theorized that natural or pharmacological antioxidants can exert a retinal neuroprotectant effect by mitigating the free radicals generated from ectopic aerobic metabolism in the outer segment disks and prevent damage to the OS [84]. Our study supports this hypothesis, as top neuroprotective agents identified from our analysis including polyphenols (e.g., curcumin) and anti-diabetics (e.g., metformin) inhibit ROI production via modulation of ectopic ATP synthase in light-exposed bovine rod OS [85,86].

### Limitations

Neuroprotection involves a complex interplay of genetics, environmental factors, diet, and lifestyle. The approach employed in this article has certain drawbacks because we only consider existing relevant genes to predict medicines. First, there are inherent errors in datasets such as STITCH and CTD including false positives and proposed chemical-gene associations which come from activity of downstream molecules to the original chemical interactions; thus, it is difficult to determine the relative value of each chemical-gene association. Second, genomics-based predictions rely on previously published literature to detect drug-gene interactions, and so there can be inequity depending on the amount of literature available for a given condition. As a result, prevalent conditions such as cancer or diabetes that have been thoroughly studied would naturally have more chemical and gene connections documented than more recent topics such as neuroprotection. Similarly, extensively studied drugs with more known disease–chemical connections may be overrepresented in enrichment analyses compared to little-known drugs due to a lack of relevant gene–disease research. Third, we did not account for positive (protective variant) or negative (risk variant) impacts of the genes. Lastly, there are predicted drug–gene interaction that may not apply physiologically to humans because the data came from animal models and transformed cell lines. Ultimately, this research process is similar to meta-analysis of hundreds of prior works, and it is necessary to account for changes in data standards and experimental methodologies among different studies. It is important to note that the nature of this study as a computational analysis means that our results are merely meant to guide future in vitro and in vivo preclinical studies as well as clinical studies studies. The expectation of this study is thus to guide subsequent studies to further our understanding of the therapeutic potential on retinal pathologies for the compounds we have identified.

## 4. Materials and Methods

### 4.1. Literature Search and Data Extraction

We queried the NCBI database “https://www.ncbi.nlm.nih.gov/gene/ (accessed on 24 August 2021)” to compile a comprehensive list of retinal neuroprotection genes. The collection of genes was performed according to the method described in prior studies [87,88].

We also reviewed the abstracts of initial publications and evaluated the genetic association studies of retinal neuroprotection. We limited our choices by concentrating on the publications that found substantial links between genes and retinal neuroprotection. By removing publications that showed negative or negligible relationships, the number of false-positive findings is minimized. For selected publications, we reviewed the complete texts to ensure that the content supported the results. Ethical approval was not required since this study does not involve humans or animals.

### 4.2. Discovering Potential Neuroprotection Therapeutic Targets via Enrichment Analysis

The level of associations between each candidate neuroprotection target gene and relevant drug was examined to discover potential neuroprotection target drugs based on the hypothesis that drugs will have a stronger impact on disease gene with a greater number of associations [89,90].

Thus, the compounds that are highly interacting with the curated neuroprotection genes can serve as potential therapeutic targets. The enrichment analysis can provide a list of over-represented drug compounds regarding the input genes against the chemical-gene association database.

Using the ToppGene Suite (http://toppgene.cchmc.org/), we inputted the gene dataset derived from our PubMed query into the ToppFun function to perform a gene set enrichment analysis (GSEA). ToppGene pulls 22,832 genes and a total of 77,146 drug annotations from 5 separate sources: Broad Institute CMAP down, Broad Institute CMAP Up, CTD, Drug Bank, and Stitch to construct a Pharmacome for predicting drug-gene interactions [91]. From the inputted genes, ToppGene generates a representative profile which it compares against its 22,832 test set genes to identify drugs with the strongest associations. Drug–gene interactions in our data came primarily from STITCH and CTD databases [92,93]. STITCH derives information from prior experiments, existing databases, and current literature to predict interactions between chemicals and proteins. CTD is a manually curated, literature-based database that provides toxicogenomic relationships for chemicals, genes, phenotypes, and disease.

Hypergeometric distribution with Bonferroni correction is used to determine statistical significance with the equation below.
$$P=1-\overset{k-1}{\underset{i=0}{\sum }}\frac{\left(\begin{array}{c} M\\ i \end{array}\right)\left(\begin{array}{c} N-M\\ n-i \end{array}\right)}{\left(\begin{array}{c} N\\ i \end{array}\right)}$$

For the referenced equation, *N* correlates to the total number of genes in the background distribution, *M* to the number of genes inside the distribution which are to the gene set of interest, *n* to the size of the list of genes of interest, and *k* to the number of genes within that list which are annotated to the gene set. By default, the background distribution includes all genes with annotation. To control the false discovery rate, *p*-values are also adjusted for multiple comparisons using the Benjamini–Hochberg Procedure. We examined medications having a false discovery rate adjusted *p*-value of less than 0.05 in this study.

### 4.3. Narrowing down Drugs/Chemicals Useful in Neuroprotection

All the compounds that are deleterious to human health or that lack clinical utilizations such as particulate matter, ozone, and asbestos were removed manually. Redundant compounds suggested by multiple databases were consolidated, and the compound with the higher *p*-value is retained.

### 4.4. Visualization of Networks

A drug–gene network was constructed using the manually curated list of neuroprotective genes and the filtered list of enriched drugs. Visualization of the associations between gene and drug was performed using Cytoscape with a prefuse-directed layout based on the centrality metric of edge betweenness [94]. We chose to color drug nodes with red and gene nodes with green. Node size correlated with closeness centrality hub in which the largest nodes had the highest closeness to better visualize potential hub nodes. Genes with less than three drug connections were not included in the visualization.

We also generated a gene-pathway network from the functional enrichment of the neuroprotective gene list. Using the Lynx database system, enrichment analysis was performed for Gene Ontology (GO), disease, and pathway databases [95]. Seven of the top significant pathways (FDR adjusted *p* value < 1.52 × 10^−14^) from the enrichment analysis were selected for construction of a gene-pathway network. Visualization and analysis of the network was achieved by Cytoscape again using a prefuse force directed layout on edge betweenness with nodal size reflecting closeness centrality [94]. Pathway nodes are colored in blue whereas drug nodes are colored in green.

## 5. Conclusions

Overall, our study identified drugs and molecules potentially relevant to neuroprotection. Many of these compounds such as metformin and statins have well-known pharmacodynamics and safety profiles which make them strong candidates for repurposing and further investigation. The identification of natural and pharmacologic antioxidants in our analysis also may support the hypothesis that oxidative stress contributes to the pathogenesis of retinal neurodegeneration in aging. In particular, the ectopic production of free radicals in rod photoreceptor cells implicated in various retinal disorders have been blunted by many of the top compounds identified from our analysis, including anti-diabetics such as metformin and natural polyphenols such as curcumin. While limitations exist, this model of using advanced bioinformatics tools and drug-gene association studies can help uncover novel domains of investigation and direct more targeted preclinical and clinical research. As our systems biology approach uses an unbiased method to predict therapeutic targets, this study can serve as a guide to identify future preclinical/clinical targets and repurposable drugs. The accuracy of predictive mathematical computer models of the drug-targets-relationship will only improve as more studies with multi-omics data concerning retinal neuroprotection becomes available. This study emphasizes the importance of a computational and bioinformatic approach in furthering our understanding of unsolved processes such as retinal neuroprotection, and we hope that this methodology can be applied to other complex, multifactorial conditions.

## Figures and Tables

**Figure 1 ijms-23-12648-f001:**
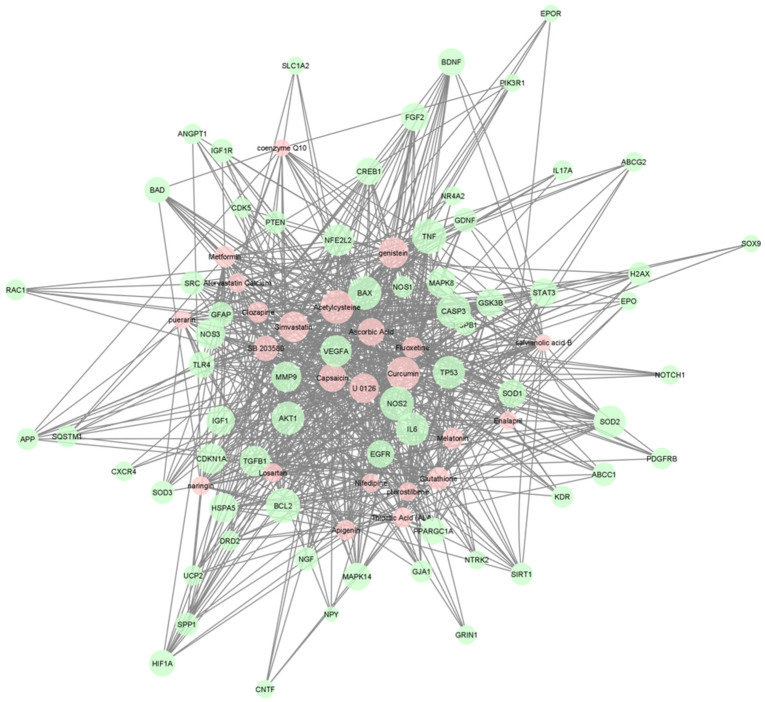
A force-directed graph of drug–gene interactions. Node size and edges are represented based on centrality metrics analysis. Drugs are in red, and genes are green with respect to color.

**Figure 2 ijms-23-12648-f002:**
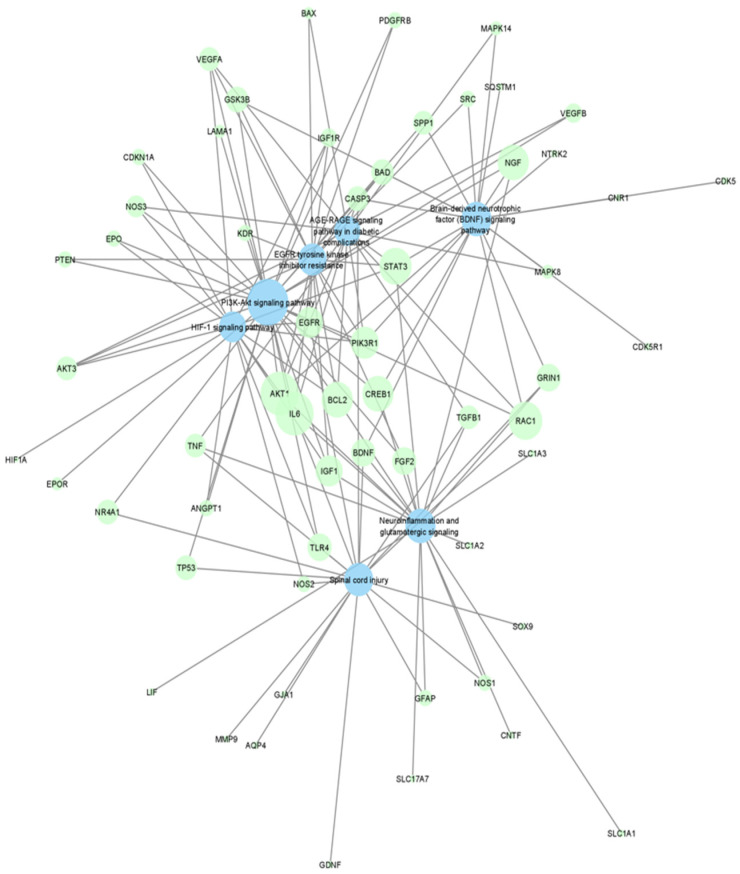
A force-directed graph of drug–gene interactions. Node size and edges are represented based on centrality metrics analysis. Pathways are in blue, and genes are green with respect to color.

**Table 1 ijms-23-12648-t001:** Filtered drugs targeting retinal neuroprotection genes predicted by the ToppGene database by order of *p*-value.

FILTERED POSITION	UNFILTERED POSITION	NAME	SOURCE	P-VALUE	Q-VALUE FDR B&H	HIT COUNT IN QUERY LIST	HIT COUNT IN GENOME
1	2	U 0126	CTD	2.51 × 10^−52^	3.64 × 10^−52^	51	444
2	3	Acetylcysteine	CTD	3.67 × 10^−55^	3.55 × 10^−51^	59	781
3	5	Simvastatin	CTD	2.72 × 10^−53^	1.58 × 10^−49^	53	581
4	7	Curcumin	CTD	1.66 × 10^−51^	6.86 × 10^−48^	58	851
5	9	Capsaicin	CTD	1.97 × 10^−49^	6.36 × 10^−46^	48	488
6	18	SB 203580	CTD	1.18 × 10^−44^	1.90 × 10^−41^	42	388
7	20	Ascorbic Acid	CTD	2.56 × 10^−41^	3.72 × 10^−38^	46	627
8	29	Genistein	Stitch	2.20 × 10^−38^	2.21 × 10^−35^	53	1117
9	43	Glutathione	CTD	3.34 × 10^−36^	2.25 × 10^−33^	35	339
10	44	Thioctic Acid	CTD	2.51 × 10^−35^	1.65 × 10^−32^	28	163
11	45	Melatonin	CTD	6.47 × 10^−35^	4.17 × 10^−32^	31	243
12	55	Nifedipine	CTD	8.36 × 10^−33^	4.41 × 10^−30^	24	112
13	61	Apigenin	CTD	2.96 × 10^−32^	1.41 × 10^−29^	28	207
14	64	Deferoxamine	CTD	5.7 × 10^−32^	2.59×10^−29^	27	186
15	72	Pterostilbene	CTD	4.13 × 10^−31^	1.66 × 10^−28^	24	130
16	84	Salvianolic acid	CTD	1.24 × 10^−31^	4.28 × 10^−28^	17	35
17	83	Fluoxetine	CTD	1.18 × 10^−30^	4.12 × 10^−28^	31	331
18	86	Puerarin	CTD	1.34 × 10^−30^	4.54 × 10^−28^	22	98
19	98	Naringin	CTD	8.18 × 10^−30^	2.42 × 10^−27^	24	146
20	101	Metformin	CTD	1.83 × 10^−29^	5.26 × 10^−27^	32	400
21	111	Atorvastatin Calcium	CTD	9.68 × 10^−29^	2.53 × 10^−26^	25	186
22	113	Losartan	CTD	1.82 × 10^−28^	4.69 × 10^−26^	24	165
23	114	Clozapine	CTD	1.92 × 10^−28^	4.89 × 10^−26^	31	390
24	127	Coenzyme Q10	CTD	1.10 × 10^−27^	2.51 × 10^−25^	17	48
25	128	Enalapril	CTD	1.38 × 10^−27^	3.14 × 10^−25^	22	131

**Table 2 ijms-23-12648-t002:** Gene ontology descriptions of genes associated with retinal neuroprotection.

ID	GO DESCRIPTION	P-VALUE	Q-VALUE FDR B&H	HIT COUNT IN QUERY LIST	HIT COUNT IN GENOME
GO:1901701	cellular response to oxygen-containing compound	6.24 × 10^−55^	3.95 × 10^−51^	78	1790
GO:0043067	regulation of programmed cell death	1.58 × 10^−53^	4.99 × 10^−50^	79	1944
GO:0009628	response to abiotic stimulus	1.04 × 10^−52^	1.65 × 10^−49^	77	1839
GO:0042981	regulation of apoptotic process	1.06 × 10^−51^	1.35 × 10^−48^	77	1897
GO:0010243	response to organonitrogen compound	4.00 × 10^−49^	4.22 × 10^−46^	71	1605
GO:1901214	regulation of neuron death	4.75 × 10^−47^	3.76 × 10^−44^	47	462
GO:0014070	response to organic cyclic compound	7.90 × 10^−47^	5.56 × 10^−44^	69	1591
GO:0048666	neuron development	2.23 × 10^−39^	6.42 × 10^−37^	64	1673
GO:0051094	positive regulation of developmental process	4.93 × 10^−39^	1.25 × 10^−36^	64	1695
GO:0042327	positive regulation of phosphorylation	1.02 × 10^−38^	2.47 × 10^−36^	55	1113
GO:0031175	neuron projection development	1.18 × 10^−37^	2.67 × 10^−35^	59	1424
GO:0070482	response to oxygen levels	1.87 × 10^−37^	3.94 × 10^−35^	45	647
GO:0031399	regulation of protein modification process	3.24 × 10^−37^	6.61 × 10^−35^	66	1976
GO:0051247	positive regulation of protein metabolic process	4.57 × 10^−37^	9.05 × 10^−35^	64	1827
GO:0009611	response to wounding	8.65 × 10^−37^	1.66 × 10^−34^	50	919

## Data Availability

Reported data gather results from PubMed Gene (https://www.ncbi.nlm.nih.gov/gene/) and the ToppGene Suite (https://toppgene.cchmc.org/) which draws from databases including Broad Institute CMAP down, Broad Institute CMAP Up, CTD, Drug Bank, and Stitch.

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
