# Peer review of "Using Computational Drug-Gene Analysis to Identify Novel Therapeutic Candidates for Retinal Neuroprotection"

_ijms, 2022, doi:10.3390/ijms232012648_

Round 1

Reviewer 1 Report

The aim of the paper from Xie et al. was to identify genes associated with retinal  neuroprotection through enrichment analysis , performed using ToppGene to identify compounds related to the identified genes. Authors  constructed  a Pharmacome from multiple drug-gene interaction databases that predicted compounds with statistically significant associations to genes involved  in retinal neuroprotection. its main contribution is, that data   indicate that the predicted drug classes with possible benefits for retinal neuroprotection include antioxidants (N-Acetylcysteine, ascorbic acid, glutathione, alpha-lipoic acid, melatonin, coenzyme Q10), polyphenols (among which curcumin, and naringin), the anti-diabetic agent metformin, lipid-lowering agents.

The scientific question is original and well defined. The results are interpreted appropriately and conclusions are justified and supported by the results. The paper is written in an appropriate way. and the data are presented appropriately. Methods are described with sufficient detail.

A weakness of the paper is bibliography is too old. Another weak  point is that the conclusions were not adequately discussed in relation to reelvant literature data. The mitochondrial redox chain proteins as well as ATP synthase were shown to be ectopically expressed, not only in the mitochondria. In particular tis was found in the rod Outer Segments (see doi: 10.1096/fba.2019-00093.)

The work could provide an advance towards the current knowledge, if it discussed the data in light of the new findings of a source of oxidative stress inside the photoreceptor outer segments, besides the retinal mitochondria. In fact this is consistent with their data showing that polyphenols, curcumin in particular, metformin and antioxidants are the most promising neuroprotectant agents. Implying that there is in fact a free radical production in the retina (see doi: 10.3390/antiox9111133.). Discussing these facts would surely increase the Interest of the Readers: the conclusions would be interesting for the readership of the Journal, and attract a wide readership; otherwise, they would remain mere correlative observations.

English language is understandable, but needs improvement.

Author Response

We would like to thank the valuable comments and we have incorporated changed to address them. We appreciate the recent findings on ectopic oxidative stress outside the retinal mitochondria at the photoreceptor outer segments per doi: 10.1096/fba.2019-00093 in our revised discussion and conclusion. We note that the damage caused by free radicals generated via this mechanism may be reduced by polyphenols such as curcumin per 10.1111/bph.13173 and anti-diabetics such as metformin per doi: 10.3390/antiox9111133 via modulation of ectopic ATP synthase. Additional citations are added to support these tie-in’s. Results from the most recent clinical trials for top drugs (various antioxidants, metformin, statin) and their efficacy in various retinal and optic nerve diseases are also now added to the discussion. 

Reviewer 2 Report

In this manuscript, Xie et al., used a computational bioinformatics approach and ToppGene to identify compounds related to the identified genes having neuroprotection in retinal neurons or related retinal diseases. Their query of PubMed Gene identified 117 unique genes associated with neuroprotection in the retina, and concluded that Anti-diabetics, lipid-lowering med-icines, and antioxidants are among the treatments anticipated by this analysis.

Overall, this manuscript provides an effective method to screen the drugs-gene related to retinal neuro protection. However, my concerns and just like authors' description, talking about each specific diseases of retina or optic nerve, every drugs needs more in vivo research is needed on both medication delivery technologies and mathematical correlations for chemical structures and biological activities. So, the limitation for this manuscript is the listed candidate drugs need further confirmation evidences in vivo and in clinical trials.

Second, for each drugs listed in the neuro protective column in this article, authors just listed their positive supports from the literature. In the discussion, authors should also list any success or fail in clinical trials or preclinical trials for each candidate for a specific retinal or optic nerve diseases.

Author Response

We would like to thank the reviewer for the insightful comment. As the reviewer states, we do describe in our paper the limitations of this bioinformatics method in needing additional in vivo or clinical research to confirm or reject our results. However, our goal in this study is to describe a method of identifying candidates - repurposable drugs that are simple and cost-effective using publicly available data to guide research as well as expedite and enhance chances of successful outcomes. These results can then be used to direct future preclincial in vivo & ex vivo and clinical studies towards drug discovery for neuroprotection and also irremediable ocular pathologies for which neuroprotection is key such as glaucoma, retinal degeneration, retinal detachment, retinal dystrophies, age related macular degeneration, diabetic retinopathy and vein occlusions. To address the reviewer’s suggestions, we included additional in vivo and clinical studies discussing our candidate drugs’ efficacy in retinal and optic nerve diseases to affirm their potential use for retinal neuroprotection. We also added in our discussion failed human studies for many of the candidate drug classes.

Amongst antioxidants, failed clinical trials are now added and discussed for ALA in the context of diabetic retinopathy. Results of combined antioxidant therapy in human trials using ALA, Vitamin C, E, and B and AREDS formulation are also now discussed for diabetic retinopathy. Limitation of AREDS compounds for AMD patients is discussed. Future direction of combined antioxidant therapy is also now mentioned. N-acetylcysteine was described for treatment of retinitis pigmentosa. Failed trials with high-dose curcumin is now cited to lead into discussion about novel formulations with higher bioavailability. Positive results of new curcumin delivery systems are reported for uveitis, central serous chorioretinopathy, and dry AMD. 

Cohort study linking metformin use to decreased risk of NPDR is added. Clinical studies observing statin therapies for DR which showed positive results and glaucoma which showed potentially harmful results are discussed as well. 

Reviewer 3 Report

The authors used a systematic computational process to discover potential therapeutic targets for retinal disorders. This approach is interesting and helpful. However, the authors' paper PMID: 35972434 "Using Advanced Bioinformatics Tools to Identify Novel Therapeutic Candidates for Age-Related Macular Degeneration" already covers most of the methods and content (same target genes in Table 1). Antidiabetic drugs, lipid regulation, and antioxidants are potential drugs. What needs to be explained is what new ideas and specific differences are provided by this work.

Author Response

We would like to thank the reviewer for the insightful comment. In this paper, we try to expand on this novel bioinformatics computational approach of our previous paper focusing on AMD to identify new potential candidates of safe, efficacious drugs for retinal neuroprotection that could be relevant for multiple ocular disorders beyond AMD. Although, there are similarities in the top drugs between this paper and our previous paper, it is reasonable to find that top drugs like various antioxidants, antidiabetics, and statins may appear to be efficacious for both a specific retinal disorder like AMD and a broad topic like retinal neuroprotection. The purpose of a systems-based computation approach is that we take an objective look at the available genes to identify targetable drugs. From the creation of the gene lists through querying NCBI to the enrichment analysis, we achieve an unbiased method of drug discovery/repurposing using existing computational tools and publicly available data. The goal of this paper is to expand our viewpoint, and it was somewhat surprising and promising to find in our analysis that many of the drugs that could target AMD could also potentially be helpful for retinal neuroprotection. In our revisions, we discuss our top drugs in the context of clinical trials with positive and/or negative results for retinal and optic nerve disease (e.g. DR, AMD, glaucoma, retinitis pigmentosa), and note that some drugs have clinical evidence supporting efficacy in multiple pathologies. We also address the use of combined-antioxidant therapy to address oxidative stress, a key element in the pathogenesis of many retinal pathologies. Lastly, we discuss recent findings from in vivo studies that ectopic oxidative stress generated at the photoreceptor outer segments outside the retinal mitochondria are an important player in retinal degenerative disease. Recent studies suggest that ROIs generated by ectopic oxidative phosphorylation are reduced by top compounds from our analysis, mainly antidiabetics such as metformin and polyphenols such as curcumin via regulation of ectopic ATP synthase (10.1111/bph.13173 & 10.3390/antiox9111133). We therefore cite excessive free radical production by the outer retina as a key player in retinal aging and degeneration. 

Round 2

Reviewer 2 Report

No further comment

Reviewer 3 Report

I have no further comments.